# Study of the mental health status of medical personnel dealing with new coronavirus pneumonia

Jun Xing[1], Ning Sun[2]*, Jun Xu[1]*, Shuling Geng[1], Yuqian Li[1]

1 Nine ward of the orthopedic, The Second Affiliated Hospital of Harbin Medical University, Harbin, P. R. China, 2 Ningbo College of Health Sciences, Ningbo, P. R. China

☯ These authors contributed equally to this work.
* sunning_ly@126.com (NS); xujun0304@163.com (JX)

**Data Availability Statement:** All relevant data are available in the OPEN ICPSR repository: https://www.openicpsr.org/openicpsr/project/119321/version/V1/view.

## Abstract

This paper studied the relationship between personality traits and mental health conditions of medical personnel to provide a basis and reference for the implementation of targeted education on mental health. A self-report inventory, the Symptom Checklist-90 (SCL-90), was used to investigate the mental health status of 548 medical personnel dealing with the new coronavirus pneumonia in eight provinces and cities of China. The overall mean SCL-90 score and mean values of factors (somatization, obsessive-compulsive, anxiety, phobic anxiety, and psychoticism) of the medical personnel were significantly higher than in the norm group ($p < 0.05$), while their average interpersonal sensitivity score was significantly lower ($p < 0.01$). In addition, personal factors affecting the mental health status of medical personnel were identified (all $p < 0.05$). The overall mental health status of medical personnel responding to new coronavirus pneumonia is generally higher than that of the norm group in China. The results of this study should contribute to measures to alleviate the psychological pressures on medical personnel dealing with the new coronavirus epidemic in China.

## 1. Introduction

Novel coronavirus pneumonia (NCP) is a pathogenic coronavirus often referred to as the novel coronavirus. On January 12, 2020, WHO officially named the disease Coronavirus Disease 2019 (COVID-19). The first case of COVID-19 was reported in Wuhan in China on December 12, 2019 [1–2] as causing severe acute respiratory infection (SARI). Coronaviruses are a large family of viruses known to cause illnesses such as cold and more serious diseases such as Middle East respiratory syndrome (MERS) and severe acute respiratory syndrome (SARS). COVID-19 is a new strain of coronavirus that has never been found in the human body before [3–4].

COVID-19 patients typically show symptoms such as fever, coughing, shortness of breath, and difficulty in breathing. In more severe cases, the disease can lead to pneumonia, severe acute respiratory failure, kidney failure, and even death [5]. There is as yet no specific

**Funding:** The authors received no specific funding for this work.

**Competing interests:** The authors have declared that no competing interests exist.

treatment for COVID-19. Patients mainly receive symptomatic treatment and care for the prevention of complications, while supplementary medical services also appear to be very effective for infected people. The rate of COVID-19 infections has increased rapidly in a short period of time. As of February 17, 2020, the number of confirmed cases exceeded 70,000, with 1,770 deaths [6]. In addition, the disease can cause secondary infections, which has created a huge burden and pressure for the prevention and treatment of the disease in various places. At present, more than 30,000 medical personnel from various medical teams across the country have provided support in Wuhan [7]. Though the help from these medical personnel has relieved the local pressure for medical care to save critically ill COVID-19 patients, there have been serious infections among the medical staff in Wuhan and other places in Hubei Province. At present, over 3000 medical personnel have been infected, which has greatly increased the psychological pressure they experience. In the face of the catastrophic health emergency of COVID-19, medical personnel have been affected by different kinds of subjective and objective factors and confront several mental health problems. Mental illness is a form of human stress response, an explanatory, emotional, and defensive response within the human body, and a physiological response of the human body to the impact of needs or injuries. Therefore, this study aims to analyze the psychological state of medical personnel dealing with COVID-19 and its influencing factors in order to provide an objective basis for the prevention of further transmission, interventions, and countermeasures for COVID-19.

## 2. Research subjects and methods

### 2.1 Subjects

This study adopted convenience sampling to recruit research subjects. From January 25 to February 16, 2020, 560 medical personnel from 12 hospitals in eight provinces and cities across the country were enrolled as research subjects. The 12 hospitals including the 1st Affiliated Hospital of Harbin Medical University, the 2nd Affiliated Hospital of Harbin Medical University, Changchun Central Hospital, Liaoning Provincial People's Hospital, Qilu Hospital of Shandong University, the Second Hospital of Tianjin Medical University, People's Hospital of Inner Mongolia Autonomous Region, the First Hospital of Ningbo, the Second Hospital of Ningbo, Guizhou Provincial People's Hospital, Sichuan Provincial People's Hospital, and Wuhan University People's Hospital.

Inclusion criteria: ① work experience ≥ 1 year; ② has given informed consent and agreed to participate in this study.

### 2.2 Methods

**2.2.1 Survey tools.** ① Survey of demographic characteristics of medical personnel, including 18 questions related to the following aspects: province, hospital, department, occupation, gender, age, highest education level, work experience, level of expertise, marital status, children, living status, whether you have participated in training for handling of public health emergencies, whether family members support your working on the front line against coronavirus, whether you have supported in affected areas in Hubei, designated hospitals, department of infectious diseases, fever clinics or emergency department, level of concern whether you and your family have been infected, degree of suspicion that you were infected when coronavirus-related symptoms occurred, and whether you have received medical observation recently. ② The SCL-90 self-report inventory: The SCL-90 inventory, compiled by Derogatis in 1975, includes 10 factors and a total of 90 items. It had been translated into Chinese version and used in the study [8]. Each factor reflects the symptoms and pain of a patient in a certain aspect, and the distribution of symptoms can be understood through the factor scores. The 10

factors include somatization, obsessive-compulsive, depression, anxiety, hostility, phobic anxiety, paranoid ideation, psychoticism, sleep, and diet. Each item was scored using a 5-point Likert scale, ranging *none*, *mild*, *moderate*, *moderate to severe*, to *severe*. The total score is the sum of the scores of the 90 items. In previous studies, the homogeneity reliability of the total SCL-90 scale was 0.97, and the homogeneity reliability of each sub-scale was also greater than 0.68. The test-retest reliability was greater than 0.7, and the content validity was above 0.85, which suggest sound reliability and validity [8].

**2.2.2 Survey methods and medical ethics.** The study used online questionnaires for data collection. Researchers conducted surveys upon the completion of general training. All procedures were approved by the Ethics Committee of Harbin Medical University (HRBYKD-A26). The research purpose and methods were explained to subjects to seek their cooperation. Online informed consent forms were signed by participants. They were informed that their participation was completely voluntary, and they could withdraw from the study at any time. The link of the online questionnaire was then sent to a total of 560 medical personnel via the Internet.

**2.2.3 Statistical methods.**

1. In this study, data analysis was performed after logical checks using the statistical software SPSS 22.0. A *p*-value less than 0.05 ($p < 0.05$) was considered statistically significant.

2. Statistical description: The mean value, standard deviation, and frequency were used to describe the demographic data of medical personnel, while the mean value and standard deviation were used to describe the scores for the mental health status of medical personnel responding to COVID-19.

3. Statistical inference: Multivariate linear regression was adopted to analyze the impact of the demographic data of medical personnel on their mental health status.

## 3. Analysis

### 3.1 General information of medical personnel

A total of 560 questionnaires were distributed, and 548 valid questionnaires were recovered, for an effective recovery rate of 97.90%. These 548 questionnaires were completed by medical personnel from eight provinces and cities in China, namely Heilongjiang, Liaoning, Jilin, Inner Mongolia, Tianjin, Sichuan, Shanxi, and Shandong. Details of the respondents' personal information are given in Table 1.

### 3.2 SCL-90 factor scores of medical personnel compared with national norms [9]

The overall average of SCL-90 and mean values of factors (somatization, obsessive-compulsive, anxiety, phobic anxiety, and psychoticism) of medical personnel were significantly higher than that of the norm group ($p < 0.05$ or $p < 0.01$), while the average score of the interpersonal sensitivity factor of medical personnel was significantly lower than that of the norm group ($p < 0.01$). Details are provided in Table 2.

### 3.3 The influencing factors of the psychological status of medical personnel

Stepwise linear regression was performed using the total score of mental health status as the dependent variable and 17 items of personal information as independent variables. The 17 items include: hospital, department, occupation, gender, age, highest education level, work

**Table 1. Demographic data of medical personnel (*n* = 548).**

| Item | Number of cases | Composition ratio (%) |
|---|---|---|
| **Class of hospital** | | |
| Class IIIA | 469 | 85.58 |
| Class IIIB | 20 | 3.65 |
| Class IIA | 41 | 7.48 |
| Class IIB | 18 | 3.28 |
| **Hospital department** | | |
| Department of Internal Medicine | 95 | 17.34 |
| Department of Surgery | 138 | 25.18 |
| Department of Obstetrics and Gynecology | 12 | 2.19 |
| Department of Pediatrics | 7 | 1.28 |
| Emergency Department | 47 | 8.58 |
| Department of Critical Care | 121 | 22.08 |
| Outpatient Department | 17 | 3.1 |
| Operating Rooms | 10 | 1.82 |
| Others | 101 | 18.43 |
| **Occupation** | | |
| Doctor | 137 | 25.00 |
| Nurse | 411 | 75.00 |
| **Gender** | | |
| Male | 153 | 27.92 |
| Female | 395 | 72.08 |
| **Age group (years)** | | |
| Under 25 years old | 46 | 8.39 |
| 26–35 years old | 290 | 52.92 |
| 36–45 years old | 124 | 22.63 |
| 45 years and older | 88 | 16.06 |
| **Highest education level** | | |
| Technical secondary school | 10 | 1.82 |
| College | 62 | 11.31 |
| Undergraduate | 381 | 69.53 |
| Master's degree and above | 95 | 17.34 |
| **Work experience** | | |
| 5 years and below | 96 | 17.52 |
| 6–10 years | 189 | 34.49 |
| 11–15 years | 99 | 18.07 |
| 16–20 years | 55 | 10.04 |
| Over 20 years | 109 | 19.89 |
| **Level of expertise** | | |
| Entry-level | 231 | 42.15 |
| Mid-level | 209 | 38.14 |
| Senior-level | 108 | 19.71 |
| **Marital status** | | |
| Single | 141 | 25.73 |
| Married | 390 | 71.17 |
| Divorced | 17 | 3.1 |
| **Any children** | | |

(*Continued*)

**Table 1.** (Continued)

| Item | Number of cases | Composition ratio (%) |
|---|---|---|
| Yes | 357 | 65.15 |
| No | 191 | 34.85 |
| **Current living situation** | | |
| Live alone | 107 | 19.53 |
| Live with family | 417 | 76.09 |
| Live in shared accommodation | 24 | 4.38 |
| **Have you participated in training for public health emergencies?** | | |
| Yes | 317 | 57.85 |
| No | 231 | 42.15 |
| **Does your family support your working on the front line?** | | |
| Yes | 444 | 81.02 |
| No | 104 | 18.98 |
| **Have you supported in affected areas in Hubei, designated hospitals, or other places?** | | |
| Infected areas in Wuhan and Hubei | 41 | 7.48 |
| COVID-19 designated hospitals | 34 | 6.20 |
| Department of Infectious Diseases | 7 | 1.28 |
| Fever Clinics | 34 | 6.20 |
| Emergency Department | 34 | 6.20 |
| None | 398 | 72.63 |
| **Level of concern whether you and your family have been infected** | | |
| Severe | 110 | 20.07 |
| Moderate | 172 | 31.39 |
| Mild | 196 | 35.77 |
| No | 70 | 12.77 |
| **Degree of suspicion that you were infected when the novel coronavirus-related symptoms occurred** | | 18.1 |
| Severe | 28 | 5.11 |
| Moderate | 68 | 12.41 |
| Mild | 131 | 23.91 |
| No | 321 | 58.58 |
| **Have you received medical observation recently?** | | |
| Yes | 51 | 9.31 |
| No | 497 | 90.69 |

experience, level of expertise, marital status, any children, living status, whether you have participated in training for handling of public health emergencies, whether family members support your working on the front line against coronavirus, whether you have supported in affected areas in Hubei, designated hospitals, department of infectious diseases, fever clinics or emergency department, level of concern whether you and your family have been infected, degree of suspicion that you were infected when coronavirus-related symptoms occurred, and whether you have received medical observation recently. The α-values for importing and exporting a variable in the regression equation were set to 0.10 and 0.15, respectively. Factors affecting the mental health and status of medical personnel based on their significance from high to low are: the degree of suspicion that they were infected when the novel coronavirus-related symptoms occurred, the level of concern whether they and their family members have

**Table 2. Comparison of SCL-90 factor scores between medical personnel and the national norm group (X±S).**

| Factor | Medical personnel (*n* = 548) | National norm (*n* = 1388) | Percentage[#] (%) | *T*-value | *P*-value |
|---|---|---|---|---|---|
| Somatization | 1.46±0.72 | 1.37±0.48 | 33.02 | 2.984 | 0.003* |
| Obsessive-Compulsive | 1.75±0.88 | 1.62±0.58 | 37.23 | 3.590 | <0.0001** |
| Interpersonal sensitivity | 1.51±0.78 | 1.65±0.61 | | 4.251 | <0.0001** |
| Depression | 1.53±0.79 | 1.50±0.59 | 29.74 | 0.761 | 0.447 |
| Anxiety | 1.50±0.79 | 1.39±0.43 | 34.12 | 3.305 | 0.001** |
| Hostility | 1.48±0.80 | 1.46±0.55 | 33.58 | 0.582 | 0.561 |
| Phobic anxiety | 1.44±0.75 | 1.23±0.41 | 39.96 | 6.446 | <0.0001** |
| Paranoid ideation | 1.40±0.73 | 1.43±0.57 | | 0.897 | 0.370 |
| Psychoticism | 1.36±0.65 | 1.29±0.42 | 32.30 | 2.391 | 0.017* |
| Others | 1.58±0.76 | | | | |
| Overall average | 1.51±0.73 | 1.44±0.43 | 32.66 | 2.109 | 0.035* |

*$p < 0.05$;

**$p < 0.0001$

[#]percentage of medical personnel with higher SCI -90 factor scores than national normal level

been infected, age, whether they have supported in affected areas in Hubei Province, designated hospitals, and other places for the novel coronavirus, and whether family members support them working on the front line ($p < 0.05$). Details of the regression results are listed in Table 3.

## 4. Discussion and summary

### 4.1 Comparison of SCL-90 factor scores between medical personnel and the national norm group

COVID-19 is a fulminant infectious disease. As it is highly contagious, many people are frightened by it and even talk fearfully about coronavirus, which can also be observed in front-line medical staff. Li et al. reported how much people and medical staff suffer from vicarious traumatization and how this vicarious traumatization of non-front-line medical staff is more serious than that of front-line medical staff [10]. As in South and Southeast Asia countries, also in Italy, there are similar problems in medical staff due to high workload and intermittent lack of protective devices. In addition, some slight form of racism is demonstrated against health care professionals who potentially have a higher risk of being infected and between non-front-line medical staff towards front-line medical staff [11]. The results of the study have shown that the overall mean of the SCL-90 and the mean values of the factors (somatization, obsessive-

**Table 3. Results of multiple linear regression analysis of influencing factors of mental health status of medical personnel.**

| Variable | B | β | *t*-value | *p*-value |
|---|---|---|---|---|
| Constant | 14.766 | | 16.312 | <0.0001 |
| Degree of suspicion that you were infected when the novel coronavirus-related symptoms occurred | 2.959 | 0.292 | 7.268 | <0.0001 |
| Level of concern whether you and your family have been infected | 2.728 | 0.300 | 7.569 | <0.0001 |
| Age | 2.787 | 0.125 | 3.417 | 0.001 |
| Have you supported in affected areas in Hubei, designated hospitals, or other places? | 1.541 | 0.121 | 3.201 | 0.001 |
| Whether your family supports your working on the front line | 6.243 | -0.094 | -2.513 | 0.012 |

$R^2 = 0.286$; $F = 44.830s$; $p = 0.001$

compulsive, anxiety, phobic anxiety, and psychoticism) of medical personnel were significantly higher than that of the norm group ($p < 0.05$), while the average score of the interpersonal sensitivity factor of medical personnel was significantly lower than that of the norm group ($p < 0.01$). The results were partly similar with some research in Wuhan city [12]. More specifically, medical personnel are most of the people closest to COVID-19 patients, so they are at high risk of exposure to the disease. Moreover, they have a deep understanding of the dangers of COVID-19, so they are prone to anxiety and fear. The infection protection procedures for COVID-19 are highly complex and medical staff need to constantly change clothes and replace protective equipment, so they are more likely to establish obsessive-compulsive behaviors. Medical personnel, especially young medical staff, have less experience in the field and in dealing with difficulties and hardships in life. Therefore, when they suddenly encounter such sudden public health events, they tend to suffer anxiety and phobic anxiety, leading to physical and mental problems. That is why scores of the factors somatization, obsessive-compulsive, anxiety, phobic anxiety, and psychoticism were significantly higher than in the norm group. This result suggests that psychologists and team leaders should pay more attention to the anxiety, phobic anxiety, and psychoticism issues of the medical personnel in a team. The average score for the interpersonal sensitivity factor of medical personnel was significantly lower than that of the norm group. This shows that in the event of an infectious disease epidemic, the majority of medical personnel are united and have good professional strengths and qualities for self-regulation and self-protection. The results of this study show partial consistency with the studies of the mental health status of front-line medical personnel for SARS in 2003 [13].

## 4.2 Analysis of influencing factors of the mental health status of medical personnel

The results of this study have shown that the factors affecting the mental health status of medical personnel based on the significance from high to low are: the degree of suspicion that they were infected when the novel coronavirus-related symptoms occurred, the level of concern whether they and their family members have been infected, age, whether they have supported in affected areas in Hubei Province, designated hospitals, and other places for the novel coronavirus, and whether their family members support them working on the front line. Specific reasons are given in the following sections.

**4.2.1 Age.** Degree of suspicion that they were infected when the novel coronavirus-related symptoms occurred, the level of concern whether they and their family members have been infected, whether they have supported in affected areas in Hubei Province, designated hospitals, and other places for the novel coronavirus, and whether family members support them working on the front line.

COVID-19 patients are the main source of transmission of the disease. Patients with latent infection (i.e., asymptomatic infection) may also constitute a source of infection [14]. Medical personnel are in frequent close contact with patients during their treatment and care, hence the high risk of infection [15]. Among 138 patients admitted consecutively from January 1 to 28, 2020, to Zhongnan Hospital of Wuhan University, the proportion of medical personnel was as high as 29% [16]. A retrospective analysis of 1099 confirmed COVID-19 patients from 552 hospitals in 31 provinces (diagnosis as of January 29) found that the proportion of medical staff was 2.09% [17]. Therefore, medical staff are at high risk of infection and are under great psychological pressure. Suicidal cases were reported in India but also in other countries, Italy included, where two infected Italian nurses committed suicide in a period of a few days probably due to fear of spreading COVID-19 to patients. It is possible that fear and anxiety of falling sick or dying, helplessness will drive an increase in the 2020 suicide rates [18]. If they become

infected as a result of supporting affected areas in Hubei and COVID-19 designated hospitals, it will not only affect their physical and mental health but also that of their families. Therefore, with the emergence of symptoms and the increase in the level of concern, the mental health status of clinical medical staff may deteriorate. Furthermore, if their families do not support them working on the front line against the disease, then the psychological burden of the medical staff will also increase due to the resulting sense of conflict with professional ethics, resulting in further impact on their physical and mental health.

**4.2.2 Age.** We found that the higher the age, the higher the mental health score and the more psychological problems. Based on the age distribution of patients across the country, all ages have no resistance to the novel coronavirus, and a person of any age can be infected as long as virus transmission conditions are met [19]. An analysis of 4021 confirmed patients nationwide (diagnosis date as of January 26) also shows that people of all ages are generally vulnerable to the disease, of whom 71.45% are aged 30 to 65 years [19]. As people get older, the risk of exposure to the disease may increase in people with underlying illness such as asthma, diabetes, and heart diseases [20]. Therefore, older medical personnel have more psychological stress when dealing with COVID-19 patients. It is advised that older medical personnel receive psychological counseling before and during work to help them adjust their status as soon as possible. Despite all the above, this study believes that after such a "smokeless" war against the novel coronavirus, the psychological quality of medical personnel can be improved to a certain extent.

## 4.3 Research limitations and future research plans

In this study, the applicability of the results is limited by the nature of cross-sectional studies, and because of its use of convenience sampling from 12 hospitals in eight provinces and cities of China. In subsequent research in this project, a longitudinal study should be conducted that uses a wider sample and measures the mental health status of medical personnel from multiple dimensions, which can help better identify the mutual influence between demographic data and mental health status.

## 4.4 Summary

In the face of the catastrophic health emergency caused by COVID-19, medical staff have been affected by different kinds of subjective and objective factors. Their mental health problems are a form of human stress response, an explanatory, emotional, and defensive response within the human body, and a physiological response of the human body to the invasion of needs or injuries. In this special environment, their work, life, and emotions tend to be regularly abnormal. Due to the requirements for isolation and disinfection, medical personnel need to wear several layers of protection clothing. This increases the intensity of their work and requires great physical energy, causing severe hypoxia and physical symptoms such as headache and muscle soreness. Other symptoms such as obsessive-compulsive symptoms, interpersonal sensitivity, depression, anxiety, phobic anxiety, hostility, and paranoid ideation are all normal psychological reactions in the handling of emergencies and environmental stimuli. In face of a disaster, persons with good mental health will tend to actively take measures such as catharsis, transference, compensation, relaxation, humor, self-consolation, and rational response. The results of this study show that the overall mental health of medical staff is generally poor when dealing with COVID-19. Psychological tests show that people have a process of adaptation to catastrophic emergencies, from initial rejection, shock, and fear, to habituation, acceptance, and calm, to co-existence and living together, which is a regular process. In the face of such a sudden disaster as COVID-19, these psychological symptoms have manifested in both doctors

and patients. For medical personnel, it is particularly important to pay attention to mental health conditions while fulfilling their responsibilities. In future research, it is worth exploring how to strengthen the monitoring of mental health conditions of medical personnel and establish an active, systematic, and scientific psychological defense system under such special circumstances.

## Supporting information

**S1 File.**
(DOC)

**S1 File.**
(DOC)

**S1 Data.**
(SAV)

**S2 Data.**
(DOC)

## Acknowledgments

The authors would like to thank the medical personnel who participated in the study.

## Author Contributions

**Conceptualization:** Ning Sun.

**Data curation:** Shuling Geng, Yuqian Li.

**Formal analysis:** Shuling Geng.

**Investigation:** Jun Xing, Yuqian Li.

**Methodology:** Jun Xing.

**Resources:** Jun Xu.

**Validation:** Jun Xu.

**Writing – original draft:** Ning Sun, Jun Xu, Yuqian Li.

**Writing – review & editing:** Ning Sun.

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
