## [Decision Letter · Decision Letter 0]

6 Apr 2020

PONE-D-20-05925

Study of the mental health status of medical personnel dealing with new coronavirus pneumonia

PLOS ONE

Dear Dr. Sun,

Thank you for submitting your manuscript to PLOS ONE. After careful consideration, we feel that it has merit but does not fully meet PLOS ONE’s publication criteria as it currently stands. Therefore, we invite you to submit a revised version of the manuscript that addresses the points raised during the review process.

We would appreciate receiving your revised manuscript by May 5, 2020. To enhance the reproducibility of your results, we recommend that if applicable you deposit your laboratory protocols in protocols.io, where a protocol can be assigned its own identifier (DOI) such that it can be cited independently in the future. For instructions see: http://journals.plos.org/plosone/s/submission-guidelines#loc-laboratory-protocols

We look forward to receiving your revised manuscript.

Kind regards,

Stephan Doering, M.D.

Academic Editor

PLOS ONE

Journal Requirements:

2. Please include in your Methods section (or in Supplementary Information files) the participating hospitals/institutions.

3. Please include additional information regarding the survey or questionnaire used in the study and ensure that you have provided sufficient details that others could replicate the analyses. If you developed and/or translated a questionnaire as part of this study and it is not under a copyright license more restrictive than Creative Commons Attribution (CC-BY), please include a copy, in both the original language and English, as Supporting Information.

4. We note that you have reported significance probabilities of 0 in places. Since p=0 is not strictly possible, please correct this to a more appropriate limit, eg 'p<0.0001'.

"The funders had no role in study design, data collection and analysis, decision to publish, or preparation of the manuscript"

6. We note that you have indicated that data from this study are available upon request. PLOS only allows data to be available upon request if there are legal or ethical restrictions on sharing data publicly. For information on unacceptable data access restrictions, please see http://journals.plos.org/plosone/s/data-availability#loc-unacceptable-data-access-restrictions.

7. PLOS requires an ORCID iD for the corresponding author in Editorial Manager on papers submitted after December 6th, 2016. Please ensure that you have an ORCID iD and that it is validated in Editorial Manager. To do this, go to ‘Update my Information’ (in the upper left-hand corner of the main menu), and click on the Fetch/Validate link next to the ORCID field. This will take you to the ORCID site and allow you to create a new iD or authenticate a pre-existing iD in Editorial Manager. Please see the following video for instructions on linking an ORCID iD to your Editorial Manager account: https://www.youtube.com/watch?v=_xcclfuvtxQ

8. Your ethics statement must appear in the Methods section of your manuscript. If your ethics statement is written in any section besides the Methods, please move it to the Methods section and delete it from any other section. Please also ensure that your ethics statement is included in your manuscript, as the ethics section of your online submission will not be published alongside your manuscript.

Reviewer's Responses to Questions

**Comments to the Author**

1. Is the manuscript technically sound, and do the data support the conclusions?

Reviewer #1: Partly

Reviewer #2: Partly

2. Has the statistical analysis been performed appropriately and rigorously? 

Reviewer #1: Yes

Reviewer #2: Yes

3. Have the authors made all data underlying the findings in their manuscript fully available?

Reviewer #1: Yes

Reviewer #2: Yes

4. Is the manuscript presented in an intelligible fashion and written in standard English?

Reviewer #1: Yes

Reviewer #2: Yes

5. Review Comments to the Author

Reviewer #1: The manuscript investigates an interesting topic that is becoming a challenging issue worldwide. The design limitations (lack of longitudinal design) could be tolerate due to the urge to disseminate research on the COVID19 pandemic. However, some modifications are requested to make the text more readable and some theoretical and background pitfalls should be addressed.

The manuscript focuses on mental health status, and the main message is that mental health problems are a consequence of the emergency condition, and psychological strain linked to the fear of the disease plays a central role. However, more international references and literature on occupational stress, burnout and psychology of emergencies should be implemented in the introduction and in the discussion.

Could you insert references of the Chinese adaptation of the SCL 90?

Please explain in the text what does the hospital class means.

Could you provide in table 2 the number or the percentage of staff members who scored higher than the national norm cut off for each scl90 subscale and for the total score? This would give the reader a synthetic idea of the mental health status of the personnel.

Regression analysis. Could you move the description of the stepwise model in the method paragraph, including table 3? You may keep the model results, including table 4, in the results section. This would make the results clearer and more readable.

Reviewer #2: Interesting paper. Research subjects and methods are correct.

Discussion must be improved. These papers can be intersting and related with your paper:

Montemurro N.

The emotional impact of COVID-19: from medical staff to common people.

Brain Behav Immun. 2020 Mar 30. pii: S0889-1591(20)30411-6. doi: 10.1016/j.bbi.2020.03.032. [Epub ahead of print]

Li Z, Ge J, Yang M et al.

Vicarious traumatization in the general public, members, and non-members of medical teams aiding in COVID-19 control.

Brain Behav Immun. 2020 Mar 10. pii: S0889-1591(20)30309-3. doi: 10.1016/j.bbi.2020.03.007. [Epub ahead of print]

Huang J, Liu F, Teng Z et al.

Care for the psychological status of frontline medical staff fighting against COVID-19.

Clin Infect Dis. 2020 Apr 3. pii: ciaa385. doi: 10.1093/cid/ciaa385. [Epub ahead of print]

6. PLOS authors have the option to publish the peer review history of their article (what does this mean?). If published, this will include your full peer review and any attached files.

Reviewer #1: Yes: Filippo Rapisarda

Reviewer #2: No

---

## [Author Response · Author response to Decision Letter 0]

16 Apr 2020

Journal Requirements:

Answer: We have checked the link and revise the manuscript according the PLOS ONE's style.

2. Please include in your Methods section (or in Supplementary Information files) the participating hospitals/institutions.

Answer: We have added the participating hospitals in the method section as following: The 12 hospitals including the 1st Affiliated Hospital of Harbin Medical University, the 2nd Affiliated Hospital of Harbin Medical University, Changchun Central Hospital, Liaoning Provincial People's Hospital, Qilu Hospital of Shandong University, the Second Hospital of Tianjin Medical University, People's Hospital of Inner Mongolia Autonomous Region, the First Hospital of Ningbo, the Second Hospital of Ningbo, Guizhou Provincial People's Hospital, Sichuan Provincial People's Hospital, and Wuhan University People's Hospital.

3. Please include additional information regarding the survey or questionnaire used in the study and ensure that you have provided sufficient details that others could replicate the analyses. If you developed and/or translated a questionnaire as part of this study and it is not under a copyright license more restrictive than Creative Commons Attribution (CC-BY), please include a copy, in both the original language and English, as Supporting Information.

Answer: we have added the details: It had been translated into Chinese version and used in the study. Both the original language and English version have been attached as supporting information.

4. We note that you have reported significance probabilities of 0 in places. Since p=0 is not strictly possible, please correct this to a more appropriate limit, eg 'p<0.0001'.

Answer: Thank you so much for your suggestion. We have use the p<0.0001 instead of the p=0

"The funders had no role in study design, data collection and analysis, decision to publish, or preparation of the manuscript"

a. Please clarify the sources of funding (financial or material support) for your study. List the grants or organizations that supported your study, including funding received from your institution.

d. If you did not receive any funding for this study, please state: “The authors received no specific funding for this work.”

Answer: Thank your so much for your suggestion. The study did not receive any funding. So we state: d “The authors received no specific funding for this work.” We have added the statement in our cover letter. 

6. We note that you have indicated that data from this study are available upon request. PLOS only allows data to be available upon request if there are legal or ethical restrictions on sharing data publicly. For information on unacceptable data access restrictions, please see http://journals.plos.org/plosone/s/data-availability#loc-unacceptable-data-access-restrictions.

Answer: Thank you so much for your direction. We have revised the content as following: All relevant data are within the manuscript and its Supporting Information files. And upload the data in the attachment. Thanks for your suggestion.

7. PLOS requires an ORCID iD for the corresponding author in Editorial Manager on papers submitted after December 6th, 2016. Please ensure that you have an ORCID iD and that it is validated in Editorial Manager. To do this, go to ‘Update my Information’ (in the upper left-hand corner of the main menu), and click on the Fetch/Validate link next to the ORCID field. This will take you to the ORCID site and allow you to create a new iD or authenticate a pre-existing iD in Editorial Manager. Please see the following video for instructions on linking an ORCID iD to your Editorial Manager account: https://www.youtube.com/watch?v=_xcclfuvtxQ

Answer: Thank you so much for your direction. I have had the ORCID ID in the past. When I validated it in Editorial Manager according the direction. The web showed the content as following:

 But when I entered in the submit system. The web showed as following: 

So if it have been validated or what can I do next? Thanks for your directing. 

8. Your ethics statement must appear in the Methods section of your manuscript. If your ethics statement is written in any section besides the Methods, please move it to the Methods section and delete it from any other section. Please also ensure that your ethics statement is included in your manuscript, as the ethics section of your online submission will not be published alongside your manuscript.

Answer: I have deleted the ethics section after the discussion and put it in the Methods section according your request. Thank for your suggestion. 

Comments to the Author

1. Is the manuscript technically sound, and do the data support the conclusions?

Reviewer #1: Partly

Reviewer #2: Partly

Answer: We have read the content carefully. Thanks for your direction.

2. Has the statistical analysis been performed appropriately and rigorously? 

Reviewer #1: Yes

Reviewer #2: Yes

Answer: We have read the content carefully. Thanks for your direction.

3. Have the authors made all data underlying the findings in their manuscript fully available?

Reviewer #1: Yes

Reviewer #2: Yes

Answer: We have read the content carefully. Thanks for your direction.

4. Is the manuscript presented in an intelligible fashion and written in standard English?

Reviewer #1: Yes

Reviewer #2: Yes

Answer: We have read the content carefully. Thanks for your direction.

5. Review Comments to the Author

Reviewer #1: The manuscript investigates an interesting topic that is becoming a challenging issue worldwide. The design limitations (lack of longitudinal design) could be tolerate due to the urge to disseminate research on the COVID19 pandemic. However, some modifications are requested to make the text more readable and some theoretical and background pitfalls should be addressed.

Answer: We have done some modifications making the text more readable and added some backgrounds.Thanks

The manuscript focuses on mental health status, and the main message is that mental health problems are a consequence of the emergency condition, and psychological strain linked to the fear of the disease plays a central role. However, more international references and literature on occupational stress, burnout and psychology of emergencies should be implemented in the introduction and in the discussion.

Answer: Thanks for your suggestion. We have added some references relating burnout and psychology of emergencies in the discussion.

Montemurro N. The emotional impact of COVID-19: from medical staff to common people.

Brain Behav Immun. 2020 Mar 30. pii: S0889-1591(20)30411-6. doi: 10.1016/j.bbi.2020.03.032. [Epub ahead of print]

Li Z, Ge J, Yang M et al.

Vicarious traumatization in the general public, members, and non-members of medical teams aiding in COVID-19 control.

Brain Behav Immun. 2020 Mar 10. pii: S0889-1591(20)30309-3. doi: 10.1016/j.bbi.2020.03.007. [Epub ahead of print]

Huang J, Liu F, Teng Z et al.

Care for the psychological status of frontline medical staff fighting against COVID-19.

Clin Infect Dis. 2020 Apr 3. pii: ciaa385. doi: 10.1093/cid/ciaa385. [Epub ahead of print]

Goyal K, Chauhan P, Chhikara K, et al.Fear of COVID 2019: First suicidal case in India![J].Asian J Psychiatr. Published online first 27 Feb 2020. DOI: 10.1016/j.ajp.2020.101989

Could you insert references of the Chinese adaptation of the SCL 90?

Answer: Thanks for your suggestion. We have inserted the reference of the Chinese adaptation of the SCL 90 as following: It had been translated into Chinese version and used in the study[8].8.Wang YY, Jia XR, Song JQ, et al. Mental health status of medical staff during the outbreak of Coronavirus Disease 2019[J]. M edical Journal of W uhan University, Published online first 17 Mar 2020. DOI: 10. 14188/j. 1671⁃8852. 2020. 0154.

Please explain in the text what does the hospital class means.

Answer: Hospital class apply with the Chinese hierarchical hospital management. The management system divides hospitals into first, second and third levels. The division of grades is based on the number of beds: less than 100 beds are first-class hospitals; 101 beds to 500 beds are classified as second-level hospitals; More than five hundred, set as three level hospitals.

Could you provide in table 2 the number or the percentage of staff members who scored higher than the national norm cut off for each scl 90 subscale and for the total score? This would give the reader a synthetic idea of the mental health status of the personnel.

Answer: Thanks so much for your suggestion. We have added the number or the percentage of staff members who scored higher than the national norm cut off for each scl 90 subscale and for the total score in table2 according your suggestion making reader a synthetic idea of the mental health status of the personnel.

Regression analysis. Could you move the description of the stepwise model in the method paragraph, including table 3? You may keep the model results, including table 4, in the results section. This would make the results clearer and more readable.

Answer: We have delete the table3 according your suggestion making the results clearer and more readable.

Reviewer #2: Interesting paper. Research subjects and methods are correct.

Discussion must be improved. These papers can be intersting and related with your paper:

Montemurro N. The emotional impact of COVID-19: from medical staff to common people.

Brain Behav Immun. 2020 Mar 30. pii: S0889-1591(20)30411-6. doi: 10.1016/j.bbi.2020.03.032. [Epub ahead of print]

Li Z, Ge J, Yang M et al.

Vicarious traumatization in the general public, members, and non-members of medical teams aiding in COVID-19 control.

Brain Behav Immun. 2020 Mar 10. pii: S0889-1591(20)30309-3. doi: 10.1016/j.bbi.2020.03.007. [Epub ahead of print]

Huang J, Liu F, Teng Z et al.

Care for the psychological status of frontline medical staff fighting against COVID-19.

Clin Infect Dis. 2020 Apr 3. pii: ciaa385. doi: 10.1093/cid/ciaa385. [Epub ahead of print]

Answer: We have inserted the references in the discussion and reference. Thanks for your direction. 

6.PLOS authors have the option to publish the peer review history of their article (what does this mean?). If published, this will include your full peer review and any attached files.

Do you want your identity to be public for this peer review? For information about this choice, including consent withdrawal, please see our Privacy Policy.

Reviewer #1: Yes: Filippo Rapisarda

Reviewer #2: No

Answer: We have read the content carefully. Thanks for your direction. The manuscript have no figures just tables.

---

## [Decision Letter · Decision Letter 1]

30 Apr 2020

Study of the mental health status of medical personnel dealing with new coronavirus pneumonia

PONE-D-20-05925R1

Dear Dr. Sun,

We are pleased to inform you that your manuscript has been judged scientifically suitable for publication and will be formally accepted for publication once it complies with all outstanding technical requirements.

With kind regards,

Stephan Doering, M.D.

Academic Editor

PLOS ONE

Reviewers' comments:

Reviewer's Responses to Questions

**Comments to the Author**

1. If the authors have adequately addressed your comments raised in a previous round of review and you feel that this manuscript is now acceptable for publication, you may indicate that here to bypass the “Comments to the Author” section, enter your conflict of interest statement in the “Confidential to Editor” section, and submit your "Accept" recommendation.

Reviewer #2: All comments have been addressed

2. Is the manuscript technically sound, and do the data support the conclusions?

Reviewer #2: Yes

3. Has the statistical analysis been performed appropriately and rigorously? 

Reviewer #2: Yes

4. Have the authors made all data underlying the findings in their manuscript fully available?

Reviewer #2: Yes

5. Is the manuscript presented in an intelligible fashion and written in standard English?

Reviewer #2: Yes

6. Review Comments to the Author

Reviewer #2: This paper showed the relationship between personality traits and mental health conditions of medical personnel to provide a basis and reference for the implementation of targeted education on mental health. Interesting paper. Authors answered to all comments. Well done!

7. PLOS authors have the option to publish the peer review history of their article (what does this mean?). If published, this will include your full peer review and any attached files.

Reviewer #2: No

---

## [Editor Report · Acceptance letter]

4 May 2020

PONE-D-20-05925R1 

Study of the mental health status of medical personnel dealing with new coronavirus pneumonia 

Dear Dr. sun:

I am pleased to inform you that your manuscript has been deemed suitable for publication in PLOS ONE. Congratulations! Your manuscript is now with our production department. 

With kind regards,

on behalf of

Professor Stephan Doering 

Academic Editor

PLOS ONE